# The Very First Romanian Unruptured 13-Weeks Gestation Tubal Ectopic Pregnancy

**DOI:** 10.3390/medicina58091160

**Published:** 2022-08-25

**Authors:** Ciprian Ilea, Ovidiu-Dumitru Ilie, Olivia-Andreea Marcu, Irina Stoian, Bogdan Doroftei

**Affiliations:** 1Faculty of Medicine, University of Medicine and Pharmacy “Grigore T. Popa”, University Street, no 16, 700115 Iasi, Romania; 2Clinical Hospital of Obstetrics and Gynecology “Cuza Voda”, Cuza Voda Street, no 34, 700038 Iasi, Romania; 3Department of Biology, Faculty of Biology, “Alexandru Ioan Cuza” University, Carol I Avenue, no 20A, 700505 Iasi, Romania; 4Department of Preclinics, Faculty of Medicine and Pharmacy, University of Oradea, December 1 Market Street, no 10, 410068 Oradea, Romania; 5Origyn Fertility Center, Palace Street, no 3C, 700032 Iasi, Romania

**Keywords:** tubal pregnancy, ectopic pregnancy, live fetus, first trimester

## Abstract

Tubal ectopic pregnancies remain a challenging and life-threatening obstetric condition in the early stages that unavoidably lead to abortion or rupture, further reflected by the associated maternal mortality. Therefore, in the present case report, we report the experience of a 36-year-old woman who presented to our Emergency Department with a history of moderate hypogastric pain, mild vaginal bleeding, and bilateral mastalgia, symptoms that started 20 days ago after uterine curettage for a declarative eight-week pregnancy. On admission, a physical examination showed regular standard signs. The ultrasound examination revealed in the left abdominal flank a gestational sac with a live fetus corresponding to the gestational age of 13 weeks. Given the position of the gestational sac, we suspected a possible abdominal pregnancy. Independently on her human chorionic gonadotropin (hCG) of 33.980 mIU/mL and hemoglobin (Hb) of 13.4 g/dL, the exact location of the pregnancy following ultrasound was hard to establish. Magnetic resonance imaging (MRI) examination was requested, after which we suspected the diagnosis of ovarian pregnancy. Given the paraclinical and clinical context of the worsening of painful symptoms, we decided to perform an exploratory laparoscopy in the multidisciplinary team (digestive and vascular surgeon) that showed the existence of a tubal pregnancy.

## 1. Introduction

According to the Centers for Disease Control and Prevention (CDC), ectopic pregnancy is a life-threatening condition in the early stages. Per current figures, it accounts for 2% of all cases, oscillating from 1.3% to 2.4% [1]. In terms of the actual incidence, the evidence is contradictory since studies are lacking [2].

An ectopic pregnancy defines the implantation outside the endometrial cavity [3] of the fertilized ovum found in the blastocyst stage. In 70–90% of cases, it takes place in the fallopian tubes within the ampulla. However, numerous other sites were described over the years, surrounding the fimbrial, isthmic, and interstitial segments. There are also data referring to the ovary, the myometrium, the cervix, the abdomen, and cesarean (C)-section scar [4,5], with most ectopic pregnancies diagnosed between 6 and 10 weeks of gestation [6]. Circumstances that describe cases in advanced stages also exist in the literature.

Moreover, a rupture might occur between the 5th to 9th week of pregnancy in situations of ectopic pregnancy, leading to abdominal or pelvic pain, amenorrhea, and in limited scenarios, vaginal bleeding [7]. It is rare for an ectopic pregnancy to advance into the 2nd trimester without the presence of symptoms, and a proper diagnosis can avert rupture.

Therefore, this manuscript aims to further provide evidence to the literature with a rare case report of a live 13-week ectopic tubal pregnancy, the sole documented occurrence in Romania, uncomplicated at this age of gestation.

## 2. Case Presentation

### 2.1. Patient Information

A 36-year-old female (T.I.), gravida 1, para 0, presented to our Emergency Department reporting moderate hypogastric pain, mild vaginal bleeding, and bilateral mastalgia. During the interview, she declared that symptoms started 20 days ago, despite her medical record without registration. On admission, a physical examination showed typical vital signs.

### 2.2. Clinical History

Retrospectively, she had amenorrhea for eight weeks with a positive pregnancy test result but decided to follow an elective curettage in another medical center. She stated that, before the curettage, she did not undergo a pelvic ultrasound examination.

### 2.3. Diagnostic Assessment and Investigations

The transvaginal pelvic ultrasound examination showed, in the left abdominal flank, a gestational sac with a live fetus corresponding to the gestational age of 13 weeks and an empty uterine cavity with no fluid in the pouch of Douglas (Figure 1, Figure 2 and Figure 3). Given the position of the gestational sac, we suspected a possible abdominal pregnancy, but the exact anatomical location of the pregnancy following ultrasound was hard to establish. The physical examination of the breast and ultrasound excluded noncylic mastalgia. The patient’s serum hCG was 33.980 mIU/mL with no signs of anemia, having a Hb of 13.4 g/dL. The results of the other paraclinical tests (blood and urine biochemistry) were within normal limits.

We conducted the MRI exam that showed a suspicion of ovarian pregnancy according to the description: on the left ovarian topography, there is a suggestive aspect for the gestational sac, inside which a living fetus was seen (spontaneous movements during the examination), and a placenta developed at the level of the lower wall; the gestational sac with global dimensions of ~53/64/65 mm (a-p/t/c-c) and with localization in the front of external iliac vascular bundles; on the right side, the gestational sac comes into contact with the sigmoid, without signs of invasion; venous dilatations of the utero–ovarian plexus developed perilesionally around the formation; uterus in anteroversion/anteroflexion without expansive formations; the linear endometrium (5 mm), empty uterine cavity, cervix without expansive formations; right ovary with normal follicular appearance; no free fluid in the abdominal cavity; no pelvic lymphadenopathy (Figure 4a–c).

Given the paraclinical and clinical context—the uncertainty of the positive diagnosis of the exact anatomical location of the pregnancy and accentuation of the abdominal pain—we decided to perform an exploratory laparoscopy in the multidisciplinary team (digestive and vascular surgeons) after obtaining informed consent. We actually found that it was a tubal pregnancy localized in the intestinal portion of the fallopian tubes, not an ovarian pregnancy (according to MRI) or abdominal pregnancy (according to ultrasound) (Figure 5a,b). The right ovary, fallopian tube, and the left ovary looked normal. In this context, we decided to perform a left salpingectomy (Figure 6). Her postoperative outcome was favorable, and the serum hCG levels decreased to <50 mIU/mL on the fourth day after surgery. Following the salpingectomy, the specimen (fetus and fetal annexes) were sent for anatomopathological examination (Figure 7a,b).

## 3. Discussion

Ectopic pregnancy is an obstetric first-trimester pregnancy complication [8] with a vast repertoire of locations (ampulla—70% [5], isthmus—12%, fimbria—11.1%, and interstitium—2.4%) [9]. The estimated prevalence oscillates at around 18%, while morbidity and mortality accounting for 9% and 13% of all related deaths [1]. The cases of ectopic pregnancy reaching the second trimester are rare [2].

A tubal pregnancy may emerge to a most symptomatic phase as a consequence of the lack of submucosal layer within the fallopian tube wall. It enables ovum implantation within the muscular wall, considering that trophoblasts rapidly proliferate and erode this muscularis layer. Such a phenomenon usually causes tubal rupture and might occur at 7.2 weeks ± 2.2 with significant hemodynamic consequences. As already mentioned, in some rare cases, the fallopian tubes dilate to accommodate a pregnancy until the second or third trimester of pregnancy [9]. The possible factors responsible for this situation can be represented by the tubal structural anomalies that cause an increase in the elasticity of the fallopian tube and the abnormalities of the trophoblastic invasion that does not penetrate the entire tubal wall. This argument emphasizes the risk of missing the early diagnosis of ectopic pregnancy. On two previous occasions from the literature, the authors reported ectopic pregnancies at an advanced gestational age [10,11]. In our case, the fact that a standard pelvic ultrasound (with its limitations) was not performed to locate the pregnancy before the curettage increased the risk of not diagnosing the tubal pregnancy, with possibly important implications for the final diagnosis.

Three distinct management procedures are currently applied to target an ectopic pregnancy. Thus, clear documentation is mandatory considering the fulminant attendance to an outpatient department for a proper diagnosis. It is imperative to remember the threats since the correct method relies on the ongoing examination based on a series of clinical factors [12]. In the case of a ruptured ectopic pregnancy, surgery is compulsory. A laparoscopy is preferred when the patient is hemodynamically stable, which is a procedure associated with shorter operative times and hospital stays reflected in the intra-operative blood loss and analgesia requirements [13,14,15]. On the other hand, a laparotomy should be provided to patients when presenting with a rupture and in a state of hypovolemic shock and compromised. A salpingectomy is reserved for cases where the contralateral tube is healthy; where the fallopian tube or the concerned fragment that contains the ectopic gestation is removed, a salpingostomy involves the removal of the ectopic pregnancy by dissecting the tube and fallopian tube, in situ, to preserve the fertility status [16]. Three teams performed systematic reviews whose objective was to report the reproductive outcomes in patients with a healthy contralateral tube, including studies evaluating the patient selection, surgical procedure, and follow-up period [17,18,19], but several manuscripts declare conflicting results [20,21].

Moreover, it is known that the chance of an intrauterine pregnancy is not increased after salpingostomy in contrast to salpingectomy, conservative surgical techniques without exposing women to significant tubal bleeding shortly post-operation, and the need for further treatment of persistent trophoblast [16] and supports current guidelines regarding the laparoscopic salpingectomy as the method of choice when there is a healthy contralateral tube [22].

As already mentioned, a laparoscopic salpingostomy should be conducted in the presence of contralateral tubal disease to preserve the fertility potential. Serum β-hCG levels following tubal bleeding are pointers, where the size of the ectopic pregnancy when >2 cm or β-hCG concentrations are >3000 IU/L or higher shortly before the surgery [23]. In such circumstances, women should undergo serial β-hCG measurements and methotrexate (MTX). Despite salpingostomy implications on costs, post-operative follow-up, and treatment of persistent trophoblast [24], it will surpass salpingectomy in terms of assisted conception avoidance [21].

The second alternative is medical treatment involving the usage of MTX [25,26,27], a folic acid antagonist associated with rapid cell division and mitosis arrest [16,28]. MTX is required when the patients are hemodynamically stable with unruptured tubal ectopic pregnancy with insignificant manifestations and diminished volume of free intraperitoneal fluid on ultrasound scan. Presently, intramuscular MTX is extensively used because of its efficiency when administered in a single dose [24,29].

Congruent with the previous aspects regarding patient suitability, several indexes such as weight and height alongside blood count correlated with other standard tests for kidney and liver functionality are needed. Although the cases are limited, the regime might cause hair loss or lead to toxicity of the bone marrow or of the liver. The most common symptoms include abdominal discomfort and bloating for approximately half a week [30].

While 14–20% of the women that underwent a single dose will need to repeat the process [31,32] due to the β-hCG concentration not dropping below 15% on day 4–7 after treatment, 10% must undergo surgery [33]. A less common approach for patients who have β-hCG levels > 5000 IU/L constitutes the direct injection of MTX into the ectopic pregnancy as a multi-dose protocol (day 1, 3, 5, and 7) and leucovorin (0.1 mg/kg on day 2, 4, 6, and 8) [34].

The last approach rotating around ectopic pregnancies is when they spontaneously resolve without any intervention via regression or tubal abortion as a conservative strategy [29]. The individual must not portray indications or symptoms of a ruptured ectopic pregnancy and be stable, with a consistent drop of serum β-hCG or progesterone and assessment of β-hCG (<1000 IU/L) [35] up to 3 times per week and ultrasonography with relatively high success rates in between [36].

Unfortunately, some results indicate a risk of recurrence of 10% in women with a known history of ectopic pregnancy and may increase to 25%. The most common risk factors are advanced maternal age (AMA), smoking, in vitro fertilization (IVF), and infertility due to previous abdominopelvic surgery or adhesions caused by infections or pelvic inflammatory disease [37,38,39]. It is possible that women who achieve pregnancy via assisted reproductive technology (ART), among which multiple embryo transfers (ETs) and tubal factor, have an increased risk of ectopic pregnancy [40].

The incidence might increase mainly because of sexually transmitted diseases (STDs) [5]. This is why patients must undergo transvaginal ultrasonography and measurement of serum β-hCG as per the guidelines issued [6]. Women using an intrauterine device (IUD) are at a lower risk of suffering an ectopic pregnancy in comparison to women not using contraception. However, in 53% of cases, an ectopic pregnancy might occur despite the presence of an IUD [41]. Implicitly, there are also numerous non-related ectopic pregnancy factors, counting C-section, oral contraceptives (COCs), or emergency contraception failure [42].

In our case, there was no associated risk factor. The anatomopathological report revealed, besides the classical representative aspects of tubal ectopic pregnancy (chorionic villi within the lumen of the tube), the presence of an important chronic inflammatory infiltrate at the level of the remaining tubal wall, possibly having to do with a history of pelvic inflammatory disease. However, it is essential to specify that the patient comes from a disadvantaged socioeconomic area with limited access to medical diagnostic services. A chronological overview of the previous case reports that report 13-week tubal ectopic pregnancies are discussed below (Table 1). Although numerous references concerning ectopic pregnancies can be found in the literature, those highlighting a 13-week tubal ectopic pregnancy are relatively limited.

Hamura et al. [46] performed a retrospective case review study over 56 months, analyzing the medical records of 73 women from Papua New Guinea. They reveal a rate of ectopic pregnancy of 6.3 per 1000 deliveries, with no maternal death, from which 85% were parous, 67% rural dwellers, and 62% with a documented history of sub-fertility, all following salpingectomy. Davenport et al. [47] conducted a retrospective cohort study between 2004 and 2018 in which they assessed 216 patients who received a single dose of intramuscular MTX (50 mg/m^2^) for the diagnosis of tubal ectopic pregnancy. Thus, aiming to investigate the time to resolution when the serum hCG < 5 IU/L, respectively, the need for rescue surgery, they noted a median time of 22 days to resolution with rescue surgery. For an hCG < 1000 IU/L, the median was 20 days, but when hCG > 2000 IU/L, the median was 34.5 days.

Based on all aspects, bedside point-of-care ultrasound (POCUS) is crucial for physicians since it reflects the time of diagnosis and patient and detects ruptures of ectopic pregnancies and ongoing abdominal bleeding. Thus, POCUS retains the sensitivity for ruptured ectopic pregnancies to detect an empty uterus, free fluid, gestational sac(s), and extrauterine masses [48].

Given the dimensions of the tubal pregnancy, in some cases, a transvaginal ultrasound (TVS) alone may not be the appropriate approach to differentiate between an abdominal and a tubal pregnancy, an MRI being necessary. However, TVS) and the assessment of β-hCG have both sensitivity and specificity compared with transabdominal ultrasound (TUS) in ectopic pregnancy diagnosis [49].

## 4. Conclusions

We presented a rare case of a 36-year-old woman with a live ectopic left tubal pregnancy corresponding to 13 weeks of gestation. To the authors’ best knowledge, this is the sole documented evolving tubal ectopic pregnancy in Romania, uncomplicated at this age. The diagnosis of such a pregnancy is an imaging challenge due to its rarity and location, and despite MRI examination, the diagnosis was certainly possible only after laparoscopy. The other peculiarity of the case is the failure to diagnose the ectopic pregnancy before performing the curettage on request. Therefore, an ectopic pregnancy diagnosis should always be ruled out in the first trimester of a pregnancy before an elective curettage. In addition, the diagnosis of a tubal pregnancy at 13 weeks of gestation using imaging remains a challenge.

## Figures and Tables

**Figure 1 medicina-58-01160-f001:**
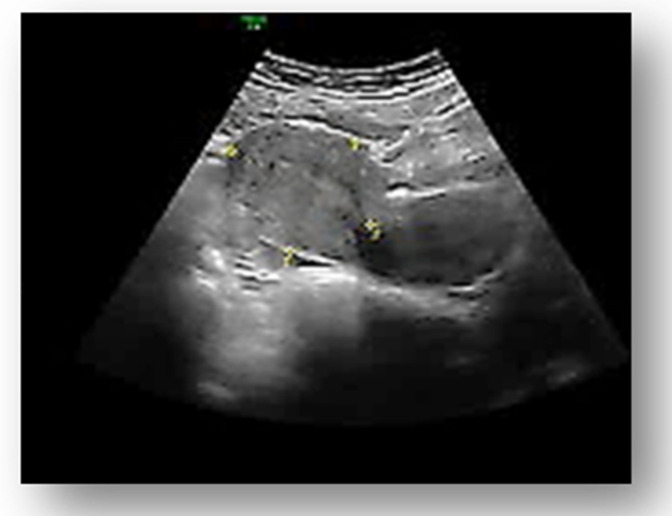
Transvaginal ultrasound from a 36-year-old woman with a left ectopic tubal pregnancy of 13 weeks of gestation. Ultrasound showed uterine body with linear endometrium and no gestational sac.

**Figure 2 medicina-58-01160-f002:**
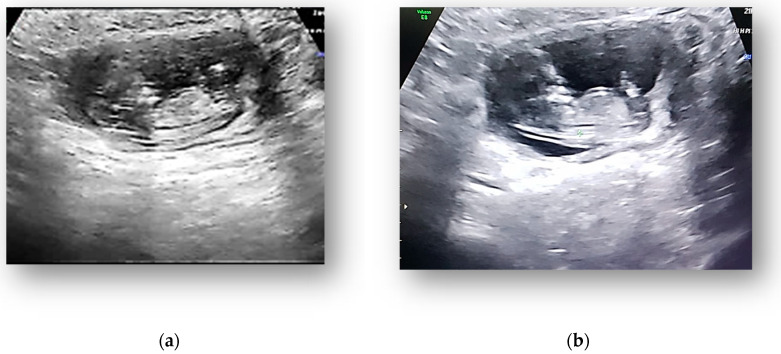
(**a**) Ultrasound showed in the left abdominal flank a gestational sac with a live fetus corresponding to the gestational age of 13 weeks. (**b**) Ultrasound showed gestational sac with fetus corresponding to 13 weeks.

**Figure 3 medicina-58-01160-f003:**
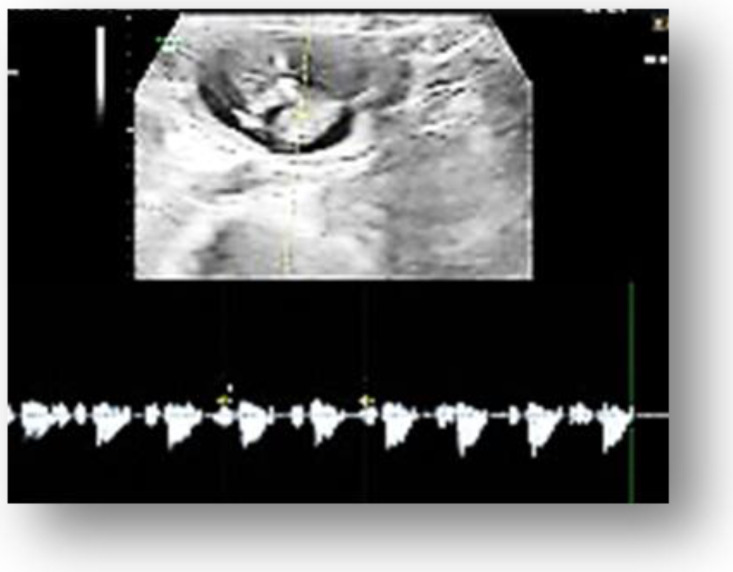
Ultrasound examination of the fetus cardiac activity.

**Figure 4 medicina-58-01160-f004:**
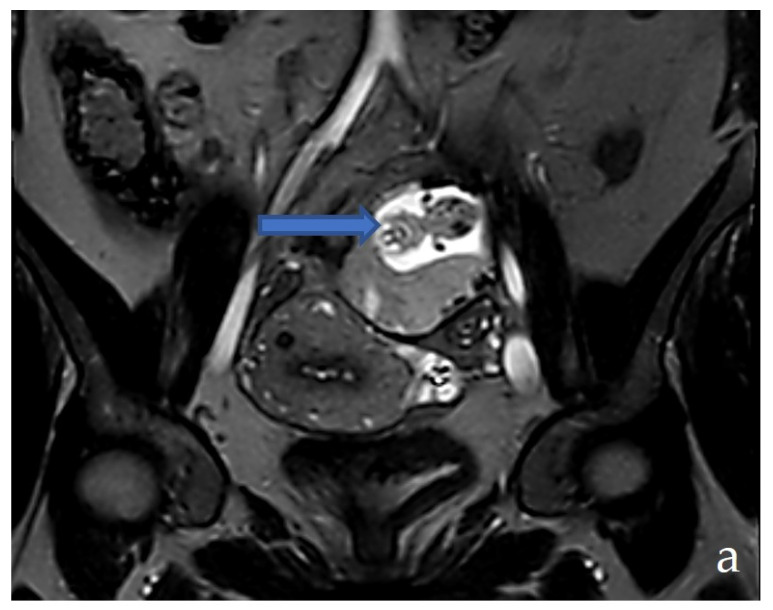
MRI examination: coronal T2-weighted (**a**), axial T2-weighted (**b**) sagittal T2-weighted (**c**)—show the ectopic pregnancy on the left ovarian topography.

**Figure 5 medicina-58-01160-f005:**
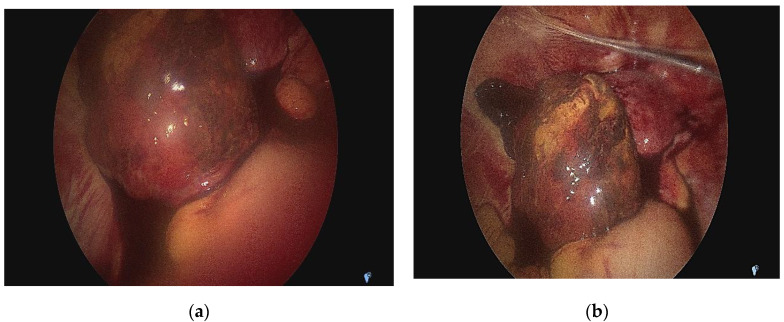
(**a**) Intra-operative image indicating the gestational sac on the left fallopian tube. (**b**) Intra-operative image indicating the gestational sac on the left fallopian tube.

**Figure 6 medicina-58-01160-f006:**
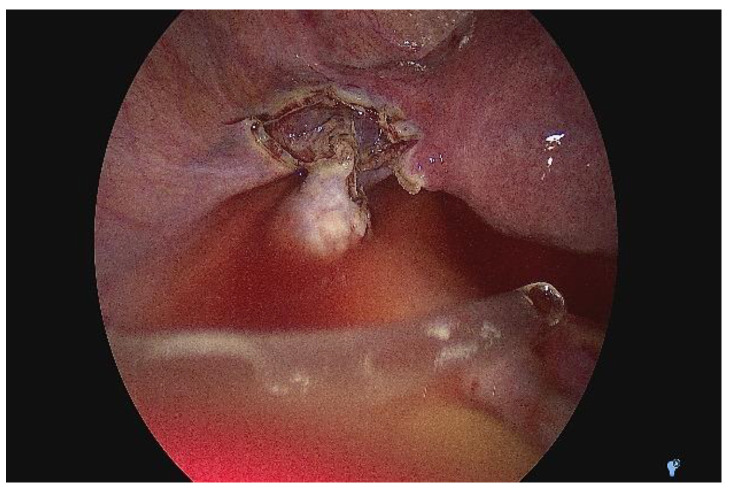
Intra-operative image after left salpingectomy.

**Figure 7 medicina-58-01160-f007:**
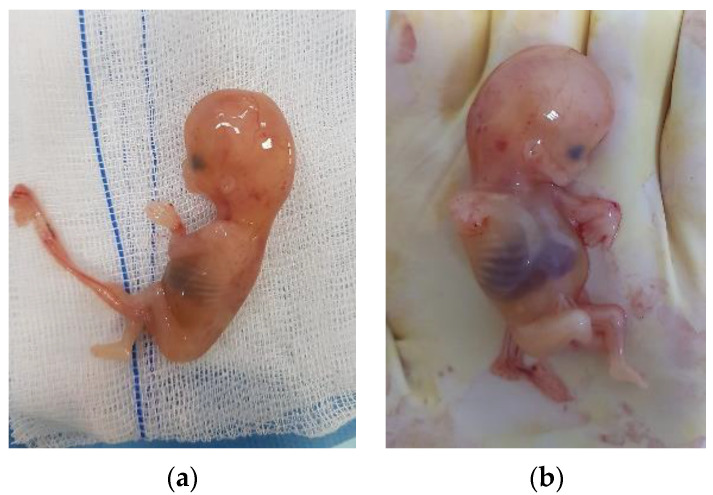
Anatomopathological specimen with the fetus after the removal of the tubal pregnancy from two distinct perspectives (**a**,**b**).

**Table 1 medicina-58-01160-t001:** A retrospective overview of 13-week tubal ectopic pregnancies.

Year of Publication	Age of Patient	Common Clinical Signs, Intervention and Weeks of Gestation	Reference
2018	31-year-old	amenorrhea for three months and one week; abdominal pain; Hb 8.5 g/dL; β-hCG 80.427, 9 mIU/mL; salpingo-oophorectomy	[43]
2019	39-year-old	abdominal pain; vaginal bleeding; Hb 8.7 g/dL; β-hCG 55.713 mIU/mL; salpingectomy	[44]
2020	38-year-old	amenorrhea for three months; abdominal pain; Hb 3.2 g/L; β-hCG 11.300 IU/mL; salpingectomy	[45]

## Data Availability

The datasets used and analyzed in this study are available from the corresponding author on reasonable request.

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
