# Peer review of "The Very First Romanian Unruptured 13-Weeks Gestation Tubal Ectopic Pregnancy"

_medicina, 2022, doi:10.3390/medicina58091160_

Round 1

Reviewer 1 Report

The article, in my opinion, must have two main objectives:firstly, to report this rare case of ectopic pregnancy; secondly,to discuss the management of the case at the light of current clinical protocols.To do that,introduction and discussion must be presented in a more clear and simple form,to reach the objectives the authors aimed to get upon. The English language needs to be completely revised. Once the correct form as well as a good English language  have been reached, the article can be considered for publication.

Author Response

Dear Reviewer #1,

We would like to thank you very much for the positive feedback, interest, and time spent reviewing our manuscript. Per your instructions, we made the respective changes that can be found below:

Comments from the Reviewer: The article, in my opinion, must have two main objectives: firstly, to report this rare case of ectopic pregnancy; secondly, to discuss the management of the case at the light of current clinical protocols. To do that, introduction and discussion must be presented in a more clear and simple form, to reach the objectives the authors aimed to get upon. The English language needs to be completely revised. Once the correct form as well as a good English language  have been reached, the article can be considered for publication.

Response: Dear Reviewer, we want to thank you greatly for your consideration to have our case report published. Per your instructions, we revised the English language throughout the entire manuscript. The Introduction section has been modified, now containing only general aspects. The discussion section presents the peculiarities of this case report, management strategies in light of the current clinical protocols, and based on our case report, risk factors, and diagnosis. Aspects and arguments regarding our case report are incorporated within distinct new and old paragraphs.

Paragraphs regarding management strategies included in the revised version:

Three distinct management procedures are applied nowadays targeting an ectopic pregnancy. Thus, clear documentation is mandatory considering the fulminant attendance outpatient department for a proper diagnosis. It is imperative to remember the threats since the correct method relies on the ongoing examination based on a series of clinical factors [12].In case of a ruptured ectopic pregnancy, surgery is compulsory. Laparoscopy is preferred when the patient is hemodynamically stable, procedure associated with shorter operative times and hospital stays reflected in the intra-operative blood loss and analgesia requirements [13–15]. On the other hand, laparotomy should be committed to patients when presenting a rupture and in a state of hypovolemic shock and compromised. Whereas salpingectomy is reserved if the contralateral tube is healthy, where the fallopian tube or the concerned fragment that contains the ectopic gestation is removed, salpingostomy involves the removal of the ectopic pregnancy by dissecting the tube and fallopian tube intact in situ to preserve the fertility status [16]. Three teams performed systematic reviews whose objective was to report the reproductive outcomes in patients with a healthy contralateral tube, studies evaluating the patient selection, surgical procedure, and follow-up period [17–19], but several manuscripts declare conflicting results [20,21].

Moreover, it is known that chance of intrauterine pregnancy is not increased after salpingostomy in contrast to salpingectomy, conservative surgical techniques without exposing women to significant tubal bleeding shortly post-operation, and the need for further treatment of persistent trophoblast [16] and supports current guidelines regarding the laparoscopic salpingectomy as the method of choice when there is a healthy contralateral tube [22].

As already mentioned, a laparoscopic salpingostomy should be conducted in the presence of contralateral tubal disease to preserve the fertility potential. Serum β-hCG levels following tubal bleeding are pointers, where the size of the ectopic pregnancy is > 2 cm or β-hCG concentrations are > 3000 IU/L or higher shortly before the surgery [23]. In such circumstances, women should undergo serial β-hCG measurements and methotrexate (MTX). Despite implications due to salpingostomy on costs and post-operative follow-up and treatment of persistent trophoblast [24], it will surclass salpingectomy in terms of assisted conception avoidance [21].

The second alternative is the medical treatment involving the usage of MTX [25–27], a folic acid antagonist associated with rapid cell division and mitosis arrest [16,28]. MTX is required when the patients are hemodynamically stable with unruptured tubal ectopic pregnancy with insignificant manifestations and diminished volume of free intraperitoneal fluid on ultrasound scan. Presently, intramuscular MTX is extensively used due to its efficiency when administered in single-dose [24,29].

Congruent with the previous aspects regarding patient suitability, several indexes such as weight and height alongside blood count correlated with other standard tests for kidney and liver functionality are needed. Although the cases are limited, the regime might cause hair loss or lead to bone marrow toxicity or of the liver. The most common include abdominal discomfort and bloating for approximately half a week [30].

While 14-20% of the women that underwent a single dose will need to repeat the process [31,32] due to β-hCG concentration not dropping below 15% on day 4-7 timescale after treatment, 10% must undergo surgery[33]. A less common approach for patients that have a β-hCG levels > 5000 IU/L constitutes the direct injection of MTX into the ectopic pregnancy as a multi-dose protocol (day 1, 3, 5, and 7) and leucovorin (0.1 mg/kg on day 2, 4, 6, and 8) [34].

The last approach rotating around ectopic pregnancies is when they spontaneously resolve without any intervention via regression or tubal abortion as a conservative strategy [29]. The individual must not portray indications or symptoms of a ruptured ectopic pregnancy and be stable, with a consistent drop of serum β-hCG or progesterone and assessment of β-hCG (< 1000IU/L) [35] up to 3 times per week and ultrasonography with relatively high success rates between [36].

Revised Introduction

According to the Centers for Disease Control and Prevention (CDC), ectopic pregnancy is a life-threatening condition in the early stages. Per current figures, it accounts for  2% of all cases,  oscillating from 1.3%  to 2.4% [1].  In terms of the actual incidence, the evidence are contradictory since studies are lacking [2].

Mechanically speaking, an ectopic pregnancy defines the implantation outside the endometrial cavity [3] of the fertilized ovum found in the blastocyst stage. In 70-90% of cases, it takes place in the fallopian tubes within the ampulla. However, numerous other sites were described over the years, surrounding the fimbrial, isthmic, and interstitial segments. There is also data referring to the ovary, the myometrium, the cervix, the abdomen, and cesarean-section scar [4,5], with most ectopic pregnancies diagnosed between the 6-10 weeks of gestation [6]. Circumstances that describe cases in advanced stages also exist in the literature.        

Moreover, a rupture might occur between the 5th to 9th week of pregnancy in situations of ectopic pregnancy.  Subsequently may lead to abdominal or pelvic pain, amenorrhea, and in limited scenarios, vaginal bleeding [7]. It is rare for an ectopic pregnancy to advance in the 2nd trimester without the presence of symptoms, and a proper diagnosis can avert rupture.

Therefore, this manuscript aims to further provide evidence to the literature with a rare case report of a live 13 weeks ectopic tubal pregnancy the sole documented evolving in Romania, uncomplicated at this age of gestation.

Kind regards and all the best,

Ovidiu-Dumitru Ilie

Reviewer 2 Report

I read with a really great interests paper entitled “The Very First Romanian Unruptured 13-Weeks Gestation Tubal Ectopic Pregnancy”. The topic of this manuscript falls within the scope of “Medicina” .

I have found some minor issues which should be corrected before final publication.

The authors should add more details considering previous medical history of this patient (number of pregnancies, abortions and deliveries (potential route of past deliveries).

Did this patient had any risk factors for ectopic pregnancy? Was it her first medical visit at this pregnancy? Did she have any ultrasound assessment at this pregnancy before emergency visit?

Again, thank you for the opportunity to review this manuscript.

Author Response

Dear Reviewer #2,

We would like to thank you very much for the positive feedback, interest, and time spent reviewing our manuscript. Per your instructions, we made the respective changes that can be found below:

Comments from the Reviewer: I  read with a really great interests paper entitled “The Very First Romanian Unruptured 13-Weeks Gestation Tubal Ectopic Pregnancy”. The topic of this manuscript falls within the scope of “Medicina”.

I have found some minor issues which should be corrected before final publication.

Response: Dear Reviewer, thank you very much for your consideration to have our case report published.

Comments from the Reviewer: The authors should add more details considering previous medical history of this patient (number of pregnancies, abortions and deliveries (potential route of past deliveries).

Response: Dear Reviewer, the female patient was gravida 1, para 0, and she had no medical record.

Comments from the Reviewer: Did this patient had any risk factors for ectopic pregnancy? Was it her first medical visit at this pregnancy? Did she have any ultrasound assessment at this pregnancy before emergency visit?

Response: In our case, there was no risk factor associated based on the interview and medical history, only that symptoms started almost three weeks before the positive pregnancy test. She did not undergo an ultrasound before presenting to the Emergency department, only that she pursued elective curettage at another center where the tubal ectopic pregnancy was not diagnosed properly.

Comments from the Reviewer: Again, thank you for the opportunity to review this manuscript.

Response: Dear Reviewer, thank you once again for your feedback regarding our case report.

Kind regards and all the best,

Ovidiu-Dumitru Ilie

Reviewer 3 Report

thanks for your invitation

overall well written,

Only

In the introduction, it should be emphasized that the risks and diagnosis of Unruptured Gestation Tubal Ectopic pregnancies cause.

best regards,

Author Response

Dear Reviewer #3,

We would like to thank you very much for the positive feedback, interest, and time spent reviewing our manuscript. Per your instructions, we made the respective changes that can be found below:

Comments from the Reviewer: thanks for your invitation, overall well written.

Response: Dear Reviewer, thank you very much for your feedback.

Comments from the Reviewer: Only

In the introduction, it should be emphasized that the risks and diagnosis of Unruptured Gestation Tubal Ectopic pregnancies cause.

Response: Dear Reviewer, per instructions received by the other Reviewers, we decided to change the Introduction and Discussion section. The Introduction section has been modified, now containing only general aspects. The discussion section presents the peculiarities of this case report, management strategies in light of the current clinical protocols, and based on our case report, risk factors, causes, and diagnosis. Aspects and arguments regarding our case report are incorporated within distinct new and old paragraphs.

Comments from the Reviewer: best regards,

Kind regards and all the best,

Ovidiu-Dumitru Ilie

Round 2

Reviewer 1 Report

For all the aspects taken in account in the review of the article, and for the form the article is now presented, in my opinion it can be accepted fo r publication, considering also the revision of english language made by the Authors.All others comments and suggestions have been made in the previous space and the Authors have followed the counsels to improve the form and the content of the article.